# Feasibility and Impact of 6-Month Rowing on Arm Lymphedema, Flexibility, and Fatigue in Breast Cancer Survivors

**DOI:** 10.3390/ijerph22070987

**Published:** 2025-06-23

**Authors:** Ester Tommasini, Paolo Bruseghini, Francesca Angela Rovera, Anna Maria Grande, Christel Galvani

**Affiliations:** 1Exercise & Sport Science Laboratory, Department of Psychology, Università Cattolica del Sacro Cuore, 20162 Milan, Italy; ester.tommasini@unicatt.it (E.T.); paolo.bruseghini@unicatt.it (P.B.); 2Senology Research Center, Department of Medicine and Innovation Technology (DiMIT), University of Insubria, 21100 Varese, Italy; francesca.rovera@uninsubria.it; 3Breast Unit, Azienda Socio Sanitaria Territoriale (ASST) Sette Laghi Hospital and University of Insubria, 21100 Varese, Italy; annamaria.grande@asst-settelaghi.it

**Keywords:** breast cancer, sculling, physical activity, lymphedema, flexibility, fatigue

## Abstract

Dragon boating and rowing are reported to be safe and provide physical benefits for women with breast cancer. Sculling, characterized by a distinct biomechanical technique, may serve as a potential tool to mitigate the adverse side effects of cancer treatments. This study investigated the feasibility and impact of a 6-month integrated physical activity program in breast cancer survivors. A longitudinal intervention study was conducted involving 20 women with breast cancer (age: 55.8 ± 6.1 yrs; BMI: 24.6 ± 3.3 kg/m^2^, stages I-III; surgery performed 6 months to 20 years prior) who participated in a 6-month exercise program consisting of three weekly one-hour sessions of adapted physical activity, walking, and sculling, with assessments conducted at baseline, 3 months, and 6 months. Physical activity, arm lymphedema, flexibility, and fatigue were tested. The program did not lead to the development or worsening of pre-existing lymphedema. A reduction of 78.9 cm^3^ was observed in the operated limb over 6 months (*p* = 0.005). An improvement in flexibility was also observed with a 2.7 cm increase in the back scratch test for the operated limb (*p* < 0.001). However, no significant change in fatigue-related variables was recorded. This is a novel study, as sculling has not previously been investigated in the context of breast cancer rehabilitation. The findings suggested that, when integrated into a structured exercise program, sculling is not only a safe and accessible activity but also effective in promoting physical and health-related improvements, with no adverse events reported. Therefore, it should be considered as part of a comprehensive rehabilitation plan for breast cancer survivors.

## 1. Introduction

Breast cancer (BC) is the most commonly diagnosed cancer among women globally. Based on estimates from the Global Cancer Observatory (GLOBOCAN) in 2022, more than 2.3 million people worldwide were newly diagnosed with BC (11.7% of all malignant tumors). Among women, it is the leading cause of cancer deaths globally and in 157 countries for incidence and in 112 countries for mortality [1]. In Europe, about 473,000 new cases of BC (in 2022) are annually diagnosed (European Cancer Information System, ECIS) [2]. In Italy, according to data from the Italian Association of Cancer Registries (AIRTUM), over 55,000 new cases were estimated in 2023. The incidence is steadily increasing, partly due to greater longevity and the implementation of screening programs, such as mammography, which allow for early diagnosis. However, mortality is slightly decreasing thanks to advances in diagnosis, surgical therapy, radiotherapy, and targeted drug treatments [3].

The management and treatment of BC involve a multi-modality approach, depending on the disease type, stage, and progression, and consist of targeted therapies, hormonal treatment, radiation, chemotherapy, and surgery. Surgery can take a long time to recover health-related Quality of Life (QoL), due to physiological and psychological side effects [4]. The repeated administration of standard adjuvant or neoadjuvant BC therapies, either before or after surgery, may exacerbate treatment-related side effects, compounding those already associated with primary therapy [5]. With the resultant increase in life expectancy among BC patients, the long-term persistence of therapy-related side effects has become increasingly apparent. These lingering effects can significantly impair health-related QoL, and can foster increased sedentary behavior, which is often further compounded by overly cautious rest prescriptions. These prescriptions frequently stem from a lack of integration of structured exercise into standard oncological care [6]. To counteract this cycle, the implementation of appropriately tailored exercise interventions is crucial.

Physical exercise, when appropriately adapted to the clinical condition of the patient, has been shown to reduce the risk of cancer recurrence and improve overall survival. It thus serves as a cornerstone of oncological rehabilitation and a key element in the promotion of long-term health in individuals diagnosed with BC. Given its systemic benefits, exercise plays a central role in reducing breast cancer-related complications and enhancing both QoL and survival outcomes [7]. For instance, the Preferable-Effect Trial (2023) demonstrated that a structured nine-month exercise intervention reduced cancer-related fatigue and significantly improved QoL in patients with metastatic BC [8]. Regarding exercise prescription, general recommendations for cancer patients often advocate for 150 min per week of moderate-intensity aerobic activity, complemented by resistance training twice weekly [9]. The American College of Sports Medicine (ACSM) guidelines specify a minimum of three sessions per week (30 min per session) of moderate aerobic exercise, along with two weekly sessions of resistance training at intensities of 60% of 1-repetition maximum (1RM), with 8–15 repetitions per set [10].

Different recreational sports have been studied with the aim of understanding their suitability for BC survivors. The main aim was to help patients to counteract the side effects of surgical, radiotherapy, and medical treatments (such as pain, lymphedema, limited mobility, cancer-related fatigue, peripheral neuropathy, neurocognitive dysfunction, cardiovascular disease, metabolic disturbances, arthralgia, depression and anxiety, bone health, respiratory impairment, etc.) with the purpose of enhancing their health-related QoL. In general, sports have been demonstrated to be safe, since no adverse events have occurred, and feasible. Some sports have proved their efficacy as treatment support for BC: Tai Chi improved vital capacity [11]; Team Triathlon (which include Running, Swimming, and Cycling) increased aerobic capacity and improved QoL [12], Nordic Walking had a positive impact on lymphedema, physical fitness, disability, and morbid perceptions [13]; Football Fitness determined positive changes in upper-body morbidity, specifically arm lymphedema [14]; Fencing allowed improvement in QoL, functional capacities, and the reduction in fatigue and anxiety [15]; and Aqua Polo seems to reduce fatigue and improve women’s psychological and social recovery [16].

Rowing is one of the most studied water sports, encompassing diverse forms and attracting BC patients because of its low-impact nature, making it a suitable exercise for patients undergoing treatment or recovering. Rowing involves the muscles of both the upper and lower limbs, of the abdomen, and almost all the body’s musculature; it combines strength and aerobic endurance with cyclic and symmetrical movements that do not require forced positions [17]. Dragon boat (DB) is a rowing sport in which a boat is propelled by 10–20 crew members distributed on both sides of the boat. A drummer leads the crew and guides the pace. Participants paddle unilaterally with a single-bladed paddle from a seated position. DB is a form of physical activity (PA) that improves the QoL of BC survivors and reduces the symptomatology caused by the disease and its treatments [18]. However, there are other types of fixed-bench boats, such as llaüt, or felucca, both typical of the Mediterranean Sea. The crew typically consists of a coxswain and eight rowers, six of them seated in pairs, with the stroke and bow rowers each on their own benches [19,20,21]. Boats are propelled with each rower holding a single oar, and the torso, legs, and arms are all involved in propulsion. Improvements were obtained in both anthropometric and physical fitness variables [21], in all the levels of PA, and in all the dimensions of QoL [20], providing physical, psychic, and emotional benefits [19].

Two rowing modalities exist, fixed-seat rowing (FSR) and sliding-seat rowing (SSR). FSR is characterized by a more asymmetrical action of the musculature and relies more on arm and torso power, while SSR involves more comprehensive movements, allowing rowers to use their legs for a significant portion of the stroke. For this reason, SSR should be more suitable for women who have had BC [22]. To date, limited attention has been directed toward the potential benefits of sculling for BC survivors. While sweep rowing, where each athlete operates a single oar on one side of the boat, has been more commonly explored in this population, sculling presents distinct biomechanical characteristics that may offer unique advantages. In sculling, each rower utilizes two oars, one in each hand, positioned symmetrically on either side of the boat. This configuration promotes a balanced and coordinated movement pattern, with the oars operating in rhythmic synchrony to propel the boat forward. Unlike sweep rowing, where the rowing motion involves significant trunk rotation towards the oar during the catch phase and a counter-rotation at the finish, sculling maintains a more symmetrical and forward-facing posture throughout the stroke cycle. However, no studies have specifically examined the feasibility, safety, or potential benefits of sculling, such as its effects on lymphedema or fatigue, in BC survivors. These attributes and the current lack of evidence highlight the need for further investigation into the role of sculling as a potentially valuable component of survivorship care in BC rehabilitation programs, with the capacity to inform clinical exercise guidelines and expand accessible rehabilitation pathways for breast cancer survivors.

Thus, the aim of this study was to understand the feasibility and impact of a 6-month-exercise intervention, including sculling, on flexibility, arm lymphedema, and fatigue in BC survivors. Given its biomechanical characteristics, the outdoor setting, and the team-based nature of sculling, we hypothesized that the intervention would be viable and well-tolerated, and would lead to improvements in flexibility, a reduction in lymphedema, and decreased cancer-related fatigue.

## 2. Materials and Methods

### 2.1. Participants

The ERICE/TSB (Effect and efficacy of RowIng in breast CancEr survivors/The Same Boat) study involved twenty women, aged 55.8 ± 6.1 years. Participants were recruited from the Breast Unit-ASST dei Sette Laghi (Varese, Italy), under the condition of having overcome BC and meeting the following inclusion criteria: (1) aged 40–65 years; (2) in the stages of BC evolution 0-1-2-3; (3) concluded postoperative cancer treatments (chemotherapy and/or radiotherapy); and (4) received sports physician and physiatrist’s clearances for PA. Exclusion criteria were set as follows: (1) the presence of cardiovascular disease, uncontrolled hypertension, and/or diabetes; (2) had psychological or psychiatric therapy with psychoactive medications in the previous 6 months; (3) the presence of musculoskeletal disturbances that could limit participation in the exercise training program; and (4) engagement in another research project. On-going hormonal therapy did not preclude study participation.

Additionally, considering the negative impact of systemic anticancer therapies on cardiovascular health [23], all participants underwent a preparticipation resting and exercise electrocardiogram screening to assess heart function and to prevent potential cardiovascular complications during the training program.

Baseline characteristics of the study participants are presented in Table 1. All women were Caucasian and did not engage in any form of structured PA.

### 2.2. Study Design

This study was a quasi-experimental longitudinal panel design. The same participants were evaluated three times: at the beginning (T0), after 3 months (T1), and at the end (T2) of the training program, as shown in Figure 1.

Before starting the training program, a meeting was conducted to outline the study’s nature, objectives, and required commitments. This study was conducted in accordance with the ethical principles for research with human subjects of the Declaration of Helsinki [24] and was approved by the Ethics Commission “Commissione Etica per la Ricerca in Psicologia (CERPS)” of the Università Cattolica del Sacro Cuore of Milan (protocol code: 44-23). Following written informed consent and the completion of baseline assessments, eligible participants commenced the training program. All evaluations were performed again at T1 and T2 under the same conditions and at the same time of day (approximately 8 a.m. to 2 p.m.). Participants were instructed to abstain from coffee and smoking on the morning of the tests, fasting for at least two hours prior, avoiding vigorous PA in the 24 h preceding the assessments, and ensuring a minimum of 7 h of sleep the night before.

### 2.3. Training Program

The training program lasted 24 weeks in which the women carried out three weekly sessions of exercise: 1 h of sculling training, 1 h of walking, and a 1 h online adapted physical activity (APA) class. A minimum adherence rate of 75% to the program was required from participants. Therefore, attendance was monitored by trainers and kinesiologists for each session and cross-checked with participants’ self-reported activity diaries. Any physical complaints, injuries, or symptoms interfering with session participation were considered adverse events and were monitored weekly through direct feedback from participants to professionals. Additionally, participant feedback on program acceptability was regularly collected through informal discussions with rowing and exercise specialists to better assess the overall participant experience of the intervention.

The sculling sessions were carried out on the Comabbio Lake (Canottieri Corgeno, Varese, Italy) and supervised by trainers who monitored the correct execution of the tasks and the session intensity. Technical difficulty and intensity (light to moderate) were progressively increased and regulated through the participants’ subjective perception of effort. Initially, indoor training sessions were conducted using an indoor rowing tank and rowing machines, combined with exercises aimed at increasing upper and lower body muscle mass and strengthening the core. Subsequently, as participants gained confidence, they transitioned to rowing on the lake using boats designed for 2 or 4 individuals (Jole and Gig). The crew comprised 2 to 4 rowers, seated in a single line along the boat. Each of the rowers used both the oars, sitting on a sliding seat, with the torso, lower, and upper limbs all actively contributing to propulsion. The technical movement of the oars needed to be performed as a cyclic and continuous action to ensure the boat sailed evenly. Therefore, this activity carried out on the lake consistently promoted teamwork.

Moreover, in order to promote weight-bearing activity, participants were asked to independently complete one hour of walking per week. Additionally, to enhance joint flexibility, postural control, and strength, they attended one hour of APA classes per week, delivered online by kinesiologists. The sessions comprised a warm-up phase including muscle activation, mobility, and postural control exercises; a workout phase characterized by resistance circuit training; and concluded with a cool-down phase focusing on flexibility and relaxation techniques.

### 2.4. Measures

#### 2.4.1. Anthropometry

Height and weight were measured using a mechanical scale and portable stadiometer (761 and 213, Seca, Hamburg, Germany). The participants were measured without shoes and without clothing (only underwear). The body mass index (BMI) was subsequently calculated as body weight in kilograms divided by height squared in meters (kg/m^2^).

Waist circumference (WC) and hip circumference (HC) were measured with a retractable measuring tape (SECA 201, Hamburg, Germany) with participants in a standing position. WC was measured by placing the tape horizontally at the midpoint between the lowest rib and the top of the iliac crest. HC was measured at the level of the maximum circumference of the hips [25]. The average of two consecutive measurements was recorded for both circumferences.

#### 2.4.2. Physical Activity Level

The participants completed the Global Physical Activity Questionnaire (GPAQ) to assess their PA level and sedentary behavior in a typical week. This questionnaire consisted of sixteen questions covering three domains: occupational, transport-related, and leisure-time PA. From the GPAQ, we extracted data on sedentary behavior (SED, expressed in minutes per week) and on the weekly minutes of moderate-to-vigorous physical activity (MVPA) within the recreational domain [26]. This domain was selected for analysis as it represented the only area in which a change was expected following participation in the intervention program.

#### 2.4.3. Lymphedema

The traditional method, which involves comparing the circumferences of both arms using a tape measure, was used to identify the presence of arm swelling and, consequently, to assess the lymphedematous arm [27,28]. These circumferences were essential for distinguishing the size of each limb and identifying the presence of lymphedema. Measurements were taken at seven sites: metacarpus, wrist, +14 cm and +7 cm from the wrist, elbow, and −7 cm and −14 cm from the elbow. These measurements were used to calculate the total arm volume (AV_tot_) of both of the upper limbs. Subsequently, the volume was determined by assuming the arm had the shape of a truncated cone. A free online tool [29], developed by Bevilacqua and colleagues [30], was used for the calculation of AV_tot_.

Percentage edema volume (%EV) was calculated as the volume difference between the operated and contralateral arms, divided by the volume of the contralateral arm. According to the 2020 guidelines of the International Society of Lymphology, a 5–20% increase in limb volume is classified as minimal lymphedema, whereas a 20–40% increase is considered moderate lymphedema [31,32].

The Relative Volume Change (RVC) formula was used to verify potential change induced by the training program: RVC = (A2U1)/(U2A1) − 1, where A1 and A2 are AV_tot_ on the side of the treated breast at two different time points, and U1 and U2 are AV_tot_ on the opposite side [33].

#### 2.4.4. Flexibility

The Back-Scratch (BS) test was administered to evaluate flexibility. Participants, in a standing position, were asked to bend one elbow overhead and the other behind the back with the aim of bringing the fingers of both hands as close together as possible. The position was held for 3 s and the distance between the fingertips was measured using a measuring tape. The upper limb bent overhead determined the side being measured. A score of zero was assigned when the fingertips just touched. If the fingertips did not meet, the score was negative; if they overlap, the score was positive. Each side was tested three times, with a 30 s rest between trials, and the best score was recorded.

#### 2.4.5. Fatigue

To assess the participants’ cancer-related fatigue over the previous 7 days, the Italian version of the EORTC Quality of Life Questionnaire-Fatigue (EORTC QLQ-FA12) was administered. The questionnaire includes 12 items that evaluate how fatigue interferes with daily activities and social life. The questionnaire incorporates three multi-item scales physical (PFA), cognitive (CFA), and emotional (EFA) fatigue. Additionally, two single items evaluate interference with daily life (IDL) and social sequelae (SOC) [34]. All scales and single-item measures have a score range from 0 to 100, where a higher score indicates a high level of symptomatology or problems. The scores for each scale and item were calculated following the instructions provided in the EORTC QLQ-FA12 Scoring Manual [35].

### 2.5. Statistical Analysis

The total sample size (*n* = 20) was estimated through an a priori power analysis. The analysis was carried out with the G*Power software (version 3.1.9.6), assuming a univariate approach for within-subject effects. The following parameters were considered: an effect size (*f*) of 0.3, an *α* err prob of 0.05, and a power (1 − *β* err prob) of 0.80, which were used to determine the required sample size.

All analyses were performed with SPSS software (Version 29). The level of significance was set at *p* < 0.05. The effect of the training program on variables was analyzed using repeated measures ANOVA with the within-subject factor “time” (T0, T1, and T2). The Shapiro–Wilk test and Mauchly’s test of sphericity were applied to assess the assumptions of normality and sphericity, respectively [36,37]. When the assumption of sphericity was violated, the Greenhouse–Geisser correction was applied [38]. Partial eta squared (η^2^_p_) was calculated to quantify the magnitude of the effect of time. Post hoc analyses were performed where appropriate, using Bonferroni-adjusted multiple comparison tests. Cohen’s *d* was computed to estimate effect sizes, with thresholds of 0.20, 0.50, and 0.80 indicating small, medium, and large effects, respectively [39].

## 3. Results

### 3.1. Training Program

After six months of the training program, no adverse events occurred and there were no dropouts. From T0 to T1, participants attended 82% of the planned training sessions (36 sessions in total, including 12 APA, 12 walking, and 12 sculling sessions). Specifically, four out of twenty women completed ≥ 55% of the sessions, while sixteen out of twenty completed ≥ 75%. From T1 to T2, the attendance rate decreased slightly to 76% of the 36 planned sessions, with eight participants completing ≥ 55% and the remaining participants completing ≥ 75% of sessions. Participants did not self-report any physical issues that could have affected their ability to take part in the exercise program. Absences from training sessions were primarily due to other personal post-surgical commitments, such as breast reconstruction.

### 3.2. Measures

The participants’ parameters at baseline, and after 3 and 6 months of the training program, are reported in Table 2.

No significant changes in BMI or HC were observed after either 12 or 24 weeks of the training program (both *p* > 0.05). Although a significant main effect of time was observed for WC (*F*_(2,38)_ = 3.40, *p* < 0.001, η^2^_p_ = 0.15), post hoc comparisons did not identify any significant differences between time points. Moreover, the volume of MVPA and SED showed improvements, even though these did not reach statistical significance (both *p* > 0.05).

AV_tot_ showed a significant main effect for time in both operated (*F*_(2,38)_ = 10.92, *p* < 0.001, η^2^_p_ = 0.37) and non-operated limbs (*F*_(2,38)_ = 9.11, *p* < 0.001, η^2^_p_ = 0.32). Post hoc comparisons revealed a decrease in AV_tot_ from T1 to T2 (operated limbs, *p* = 0.002, *d* = 0.28; non-operated limb, *p* = 0.007, *d* = 0.23) and from T0 to T2 (operated limbs, *p* = 0.005, *d* = 0.31; non operated limb, *p* = 0.005, *d* = 0.25), as shown in Figure 2.

Consequently, the percentage of EV, which corresponded to the volume difference between the operated and contralateral arms, did not show significant changes (*p* > 0.05). However, at baseline (T0), three out of twenty participants had lymphedema classified as minimal (i.e., a 5–20% increase in limb volume). At the end of the training program (T2), none of the participants presented lymphedema. Improvements in lymphedema were consistent across participants regardless of cancer stage or the type of surgical intervention. Furthermore, considering the calculation of RVC, which accounted for changes in both the affected and unaffected arms over time, it showed on average a decrease of −0.1% after 3 months and a decrease of −0.6% after 6 months.

A significant main effect of time was found for the BS test in both the operated (*F*_(2, 38)_ = 4.71, *p* = 0.030, η^2^_p_ = 0.20) and non-operated (*F*_(2, 38)_ = 14.15, *p* < 0.001, η^2^_p_ = 0.43) limbs. Post hoc comparisons showed that flexibility significantly increased from T0 to T2 in both limbs (operated limbs, *p* < 0.001, *d* = −0.30; non-operated limb, *p* < 0.001, *d* = −0.41) and a significant increase was also recorded from T0 to T1 in the non-operated limb (*p* = 0.035, *d* = −0.25), as shown in Figure 3.

Despite small effect sizes, the improvements in lymphedema and flexibility may have meaningful implications for functional mobility and the performance of daily activities in BC survivors.

The scores for each scale and item from the EORTC QLQ-FA12 are reported in Table 3.

Repeated measures ANOVA revealed no significant main effects of time on fatigue-related variables.

## 4. Discussion

PA has recently gained attention as a complementary intervention in the management of BC in women. Various recreational sports have been explored to assess their suitability, including Tai Chi, Nordic walking, and Aqua Polo [11,13,16]. Among these, rowing has received particular attention, with several studies highlighting its potential as an appropriate exercise modality due to its low-impact nature, full-body muscle engagement, and outdoor setting [18,19,20,21,22]. Despite this growing interest in rowing-based interventions, no studies to date have specifically investigated the potential benefits of sculling for BC survivors. The present study was designed to evaluate the feasibility and effects of a six-month exercise intervention, including sculling, on arm lymphedema, flexibility, and fatigue among BC survivors. We found that the training program was safe and suitable for the participants, and that after six months, total arm volume significantly decreased and upper limb flexibility increased.

The current investigation offers the first evidence that an integrated exercise program, incorporating sculling, is both safe and feasible for BC survivors. Over the 24-week intervention, no injuries were reported, and all participants completed the program. Notably, the sample was highly heterogeneous, including both younger and older women with varying types of surgical intervention, at different stages of post-surgical recovery, and was either undergoing or had completed hormone therapy. Despite this variability, all participants were able to adhere to and complete the training protocol, underscoring the program’s broad applicability and acceptability.

No worsening was observed in the anthropometric variables measured in our study. Similar results were reported by Moro et al. [40] and Gavala-González et al. [21], who found no significant changes in weight and BMI following a 12-week DB intervention and a 12-week rowing training program, respectively. Conversely, other studies investigating the effects of 24-week rowing training programs have reported statistically significant improvements in weight [41], as well as in BMI, WC, and HC [22,41]. Although our intervention lasted the same number of weeks as those conducted by Real-Pérez et al. and Gavala-González et al., these inconsistencies may be explained by differences in training modalities (rowing alone vs. an integrated PA program), volume (150 min/week vs. 180 min/week), and intensity (vigorous vs. light to moderate).

The increase in time spent in MVPA and the decrease in SED behavior, though not statistically significant, suggested that after the intervention, participants were more active and spent less time engaged in sedentary activities throughout the week. These results are in line with those of a previous study by Gavala-González et al. [20], which reported significant improvements in PA levels and lower values for the sitting time variable as measured by the International Physical Activity Questionnaire (IPAQ), following a 12-week rowing training program. Considering that low levels of PA among cancer patients have been widely reported [42], this study provides further indication supporting sculling as a potential tool to improve PA levels.

One of the most notable findings was the significant reduction in total arm volume. The calculation of EV% suggested that the 24 weeks of sculling intervention contributed to an improvement in lymphedema. In fact, the proportion of participants with lymphedema decreased from 15% to 0%. This improvement was further supported by the negative trend observed in the RVC. These results suggested that sculling does not pose particular restrictions for BC survivors, regardless of their history of lymphedema. A recent study [40] assessed upper limb circumference before and after a 12-week DB program, nevertheless failed to detect any significant changes. It is plausible that these findings are attributable to the shorter duration of the intervention and the different types of rowing studied. Considering the physical and psychological burden associated with lymphedema, as well as the risk of exacerbation through physical exercise, the observed reduction in lymphedema prevalence among participants represents a clinically meaningful and important outcome for BC survivors.

The second major finding of this study was the improvement in upper limb flexibility observed on both sides. These results were consistent with those reported by Gavala-González et al. [22], who observed an increase in flexibility in the group practicing sliding-seat rowing, while a decrease was found in the fixed-seat rowing group. A possible explanation for this difference lies in a more asymmetrical involvement of arm muscles in the fixed-seat rowing technique, which may lead to imbalances in upper limb development. Our findings add to the existing literature on the potential benefits of symmetrical rowing techniques on upper limb flexibility. This hypothesis is further corroborated by Moro et al. [40], who observed no improvements in flexibility in women engaged in DB intervention, another sport involving asymmetrical upper body movements. Additionally, it is important to note that these gains are likely supported by the APA sessions, which included flexibility and mobility exercises specifically designed to complement the PA program and enhance the rowing technique.

Finally, this program did not result in any statistically significant changes in fatigue-related variables. However, after 6 months, improvements in emotional fatigue and social sequelae were observed, although non-significant. These results suggested that rowing, being a team sport, may have an impact on the emotional and social aspects of an individual. Fatigue is a common and long-lasting side effect of cancer treatment and is known to negatively impact QoL [43]. Physical exercise has been shown to alleviate cancer-related fatigue [10], and the DB and rowing program has already been reported to improve various dimensions of QoL in BC survivors [18,20]. A study by De Luca et al. [44] specifically investigated the effects of a 24-week combined aerobic and strength training program on fatigue, which is known to interfere with daily functioning and social life, thereby affecting overall QoL. The study, which used multiple fatigue scales through the FACIT-F questionnaire, reported a significant improvement in overall QoL. This inconsistency may be explained by differences in the instruments used to assess fatigue. As stated by Rothmund et al. in 2024 [45], variations in fatigue assessment tools can lead to different findings, due to various conceptual frameworks and item phrasing. Moreover, another possible explanation for the lack of significant improvements in cancer-related fatigue is that over 50% of the participants were undergoing hormonal therapy, which is known to be associated with persistent fatigue as a common side effect [46]. This may have limited the intervention’s effectiveness in reducing fatigue.

Adherence to exercise programs is critical for optimizing the benefits and effectiveness of interventions in women with BC. During the first three months of the exercise program, adherence exceeded 80%. However, in the latter half of the treatment, adherence declined by a few percentage points. Participation in the exercise program was not negatively impacted by injuries related to PA, and the adverse effects of therapies associated with the management of the condition were not exacerbated by the exercise regimen. Non-participation in certain training sessions was attributed to personal reasons, including the necessity of undergoing medical treatments. Nevertheless, adherence to the program remained higher than the levels reported in the literature. A 2024 literature review by Ackah et al. [47], in fact, reported that adherence to exercise therapy programs in BC patients was 63% during primary cancer treatment and 68% after primary cancer treatment. To improve adherence to an exercise program and enhance motivation, it is important to make exercise enjoyable, set realistic goals, integrate it into daily routines, and to provide consistent support and encouragement. Such motivational support may contribute to fostering engagement in exercise and PA programs during cancer treatment.

To the best of our knowledge, this study represents the first research initiative to implement an exercise therapy program for BC survivors that incorporates sculling as a form of symmetrical technical movement. Our findings highlighted the feasibility and safety of employing sculling that actively engage the operated limb, as well as the entire body. This exercise modality facilitates a coordinated motor pattern, in which the oars move in rhythmic synchrony to propel the boat forward. Notably, the absence of trunk rotation contributed to enhance vessel stability and overall safety, potentially increasing patient confidence and promoting sustained adherence to the activity. Furthermore, in contrast to the existing literature—where assessments are typically conducted only at the conclusion of the training program (often after six months)—our study implemented evaluations at three-month intervals. This more frequent monitoring enabled more precise tracking of therapeutic progress and facilitated the early identification of potential plateaus in patient development. As with any research, several limitations must be acknowledged. Firstly, this was a quasi-experimental longitudinal study and, as such, lacked a control group. Secondly, the sample exhibited substantial heterogeneity in terms of age, the type of surgical intervention, pharmacological and hormonal treatments, tumor laterality, and cancer stage. Moreover, nutritional intake and dietary patterns were not assessed, representing a potential source of confounding. Finally, the absence of specific assessments of physical fitness (e.g., cardiovascular endurance, muscular strength) represented a further limitation, as it prevented a more comprehensive evaluation of the physiological adaptations induced by the intervention. Future studies should consider integrating these variables and assessments to better elucidate their potential impact on anthropometric outcomes and overall health-related fitness.

## 5. Conclusions

Overall, the findings of this study were positive. Although statistical significance was not always achieved, the observed trends were generally favorable, and the intervention program did not lead to any worsening in the measured variables.

The results of the present study appeared encouraging; however, the limited enhancement of the data analyzed, in comparison to the existing literature, may be attributed to the inherently demanding nature of sculling. This discipline is both physically and mentally challenging, emphasizing precision and technical proficiency. Mastery in sculling requires significant time and dedicated practice to refine specific rowing techniques. A high level of balance, coordination, and the independent control of two oars is essential. Proper blade engagement, optimal body positioning, and effective oar handling are crucial skills that must be developed to propel the boat efficiently. Despite the technical and cognitive demands of sculling, which necessitate sustained mental focus and concentration, the women consistently demonstrated commitment and capability in mastering these skills and were not discouraged from learning and excelling in the discipline.

Altogether, these findings may inform clinical practice by supporting the inclusion of structured, supervised PA programs, including sculling, into survivorship care plans for breast cancer patients. Based on the promising outcomes observed, future studies should explore the effects of longer interventions and larger sample sizes to confirm and expand upon these findings.

## Figures and Tables

**Figure 1 ijerph-22-00987-f001:**
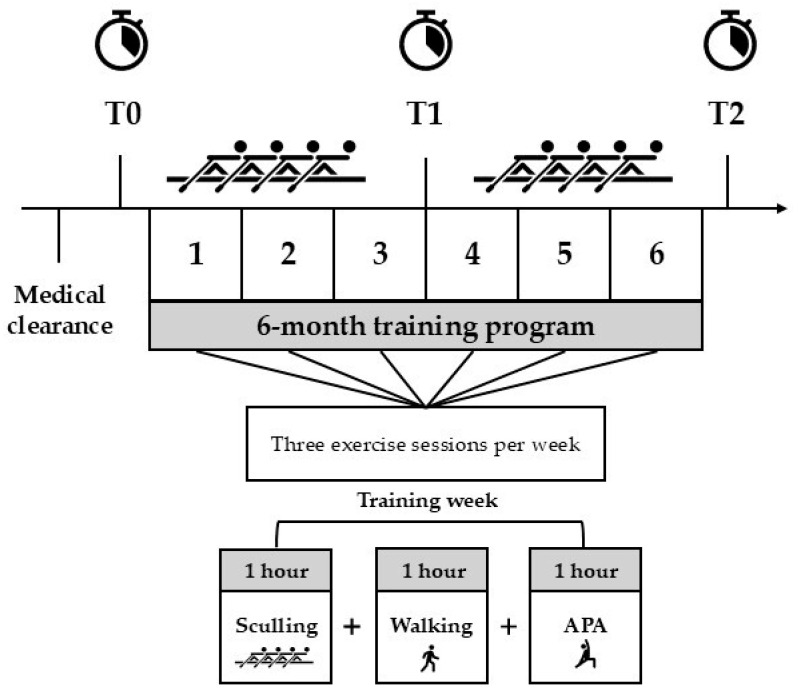
Study timeline. APA, adapted physical activity.

**Figure 2 ijerph-22-00987-f002:**
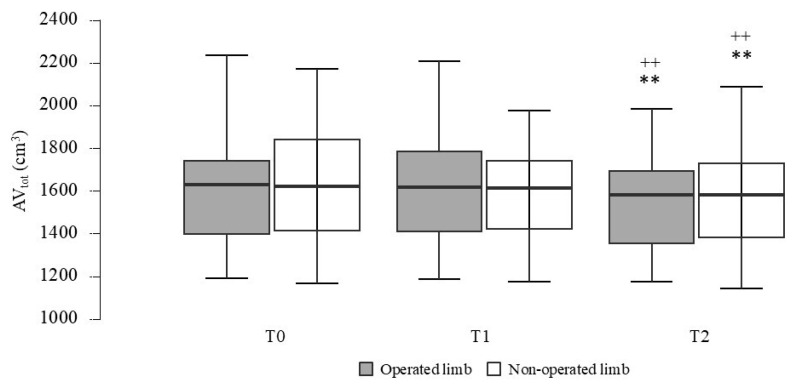
Variation in total arm volume (AV_tot_) of the operated and non-operated limbs from baseline (T0) to 3 months (T1) and 6 months (T2). ** statistically different from T0 (*p* < 0.01); ^++^ statistically different from T1 (*p* < 0.01).

**Figure 3 ijerph-22-00987-f003:**
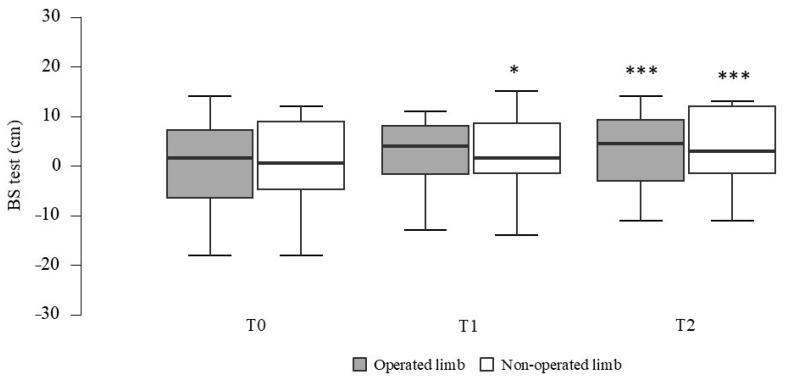
Variation in back scratch (BS) test flexibility for the operated and non-operated upper limbs at baseline (T0), after 3 months (T1), and 6 months (T2). Statistically different from T0: * *p* < 0.05; *** *p* < 0.001.

**Table 1 ijerph-22-00987-t001:** Participants’ characteristics at baseline.

Variables	Classification	Sample (*n* = 20)
Age (years)		55.8 ± 6.1
Weight (kg)		64.6 ± 9.0
Height (m)		1.62 ± 0.05
Time since surgery (months)		51.2 ± 62.6
Stage of cancer (%)		
	Stage 0	10
	Stage 1	50
	Stage 2	20
	Stage 3	15
	Unknown	5
Type of surgery (%)		
	Preservation	55
	Mastectomy	45
Cancer side (%)		
	Left	65
	Right	35
Type of treatment (%)		
	Chemotherapy	65
	Radiotherapy	70
	Hormone therapy	55

Data are presented as means ± SD.

**Table 2 ijerph-22-00987-t002:** Study variables at baseline and after 3 and 6 months of the training program.

Variables	T0	T1	T2	ANOVA-Time		
	Mean ± SD	Mean ± SD	Mean ± SD	*F*	*p*-Value	η^2^_p_
BMI (kg/m^2^)	24.6 ± 3.3	24.4 ± 3.3	24.5 ± 3.1	0.35 ^†^	0.642	-
WC (cm)	83.4 ± 9.4	84.8 ± 10.1	84.8 ± 10.1	3.40	0.044	0.15
HC (cm)	101.8 ± 8.1	100.7 ± 7.3	100.3 ± 7.8	2.79 ^†^	0.09	-
MVPA (min/week)	297 ± 351	359 ± 275	354 ± 262	0.45 ^†^	0.542	-
SED (min/week)	279 ± 153	270 ± 121	240 ± 143	0.90 ^†^	0.338	-
AV_tot_—operated limb (cm^3^)	1616 ± 274	1607 ± 259	1537 ± 218	10.91	<0.001	0.37
AV_tot_—non-operated limb (cm^3^)	1617 ± 266	1612 ± 263	1553 ± 244	9.10	<0.001	0.32
EV (%)	0.1 ± 4.6	−0.2 ± 3.4	−0.8 ± 4.2	0.29	0.752	-
BS test—operated limb (cm)	0.2 ± 9.5	1.4 ± 9.9	2.9 ± 7.9	4.71 ^†^	0.030	0.20
BS test—non-operated limb (cm)	0.2 ± 9.1	2.2 ± 8.0	3.5 ± 7.6	14.15	<0.001	0.43

Data are presented as means ± SD. BMI, body mass index; WC, waist circumference; HC, hip circumference; MVPA, moderate-to-vigorous physical activity; SED, sedentary behavior; AV_tot_, total arm volume; EV, edema volume; BS, back scratch. ^†^ Greenhouse–Geisser correction applied due to a violation of sphericity (Mauchly’s test, *p* < 0.05).

**Table 3 ijerph-22-00987-t003:** Participants’ scores from the EORTC QLQ-FA12 at baseline and after 3 and 6 months of the training program.

Variables	T0	T1	T2	ANOVA-Time		
	Mean ± SD	Mean ± SD	Mean ± SD	F	*p*-Value	η^2^_p_
PFA	24.0 ± 18.8	18.7 ± 12.7	23.3 ± 22.0	0.63	0.540	-
EFA	22.2 ± 24.7	12.8 ± 15.0	12.8 ± 17.8	2.36 ^†^	0.125	-
CFA	14.2 ± 17.3	16.7 ± 15.3	11.7 ± 21.0	0.84	0.441	-
IDL	21.7 ± 22.4	20.0 ± 22.7	20.0 ± 25.1	0.04 ^†^	0.916	-
SOC	25.0 ± 21.3	13.3 ± 19.9	15.0 ± 27.5	2.75	0.077	-

Data are presented as means ± SD. PFA, physical fatigue; EFA, emotional fatigue; CFA, cognitive fatigue; IDL, interference with daily life; SOC, social sequelae. ^†^ Greenhouse–Geisser correction applied due to a violation of sphericity (Mauchly’s test, *p* < 0.05).

## Data Availability

Data will be made available upon reasonable request.

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
