# Peer review of "Feasibility and Impact of 6-Month Rowing on Arm Lymphedema, Flexibility, and Fatigue in Breast Cancer Survivors"

_ijerph, 2025, doi:10.3390/ijerph22070987_

Round 1
Reviewer 1 Report
Comments and Suggestions for Authors
This study is a valuable contribution to exercise and sports science, involving breast cancer patients over a 6-month period. However, the study title is given as ‘Feasibility and impact of 6-month rowing on physical activity, arm lymphedema, flexibility, and fatigue in breast cancer survivors’. Similarly, in line 121, the study objective is stated as ‘Thus, the aim of this study was to understand the feasibility and impact of a 6-month exercise intervention, including sculling, on PA, flexibility, arm lymphedema, and fatigue in BC survivors.’ It is normal to investigate the effects of a 6-month rowing programme on lymphedema, flexibility, and fatigue, and the main results of the study are as expected, relating to these variables. However, what effect did the researchers expect such a study to have on participants' physical activity levels? What effect was expected from adding 1 hour of walking per week and 1 hour of adapted exercise per week to the rowing programme on physical activity levels? There is no intervention to increase participants' physical activity levels. If the researchers expected an increase in physical activity levels, which would mean that participants would engage in additional physical activity, this could have masked the effects of the rowing exercise. However, there is no change in physical activity levels. This point somewhat undermines a bit the design of the study. My recommendation is to remove the phrase ‘physical activity’ from the title and the objective section (line 121).
In addition, evaluating the effects of the 6-month rowing exercise training on participants' related physical fitness (cardiovascular endurance, strength, etc.) would make the study more comprehensive. This could also be mentioned in the limitations section.
Author Response
This study is a valuable contribution to exercise and sports science, involving breast cancer patients over a 6-month period. However, the study title is given as ‘Feasibility and impact of 6-month rowing on physical activity, arm lymphedema, flexibility, and fatigue in breast cancer survivors’. Similarly, in line 121, the study objective is stated as ‘Thus, the aim of this study was to understand the feasibility and impact of a 6-month exercise intervention, including sculling, on PA, flexibility, arm lymphedema, and fatigue in BC survivors.’ It is normal to investigate the effects of a 6-month rowing programme on lymphedema, flexibility, and fatigue, and the main results of the study are as expected, relating to these variables. However, what effect did the researchers expect such a study to have on participants' physical activity levels? What effect was expected from adding 1 hour of walking per week and 1 hour of adapted exercise per week to the rowing programme on physical activity levels? There is no intervention to increase participants' physical activity levels. If the researchers expected an increase in physical activity levels, which would mean that participants would engage in additional physical activity, this could have masked the effects of the rowing exercise. However, there is no change in physical activity levels. This point somewhat undermines a bit the design of the study. My recommendation is to remove the phrase ‘physical activity’ from the title and the objective section (line 121).
In addition, evaluating the effects of the 6-month rowing exercise training on participants' related physical fitness (cardiovascular endurance, strength, etc.) would make the study more comprehensive. This could also be mentioned in the limitations section.
Thank you very much for your insightful comment and for highlighting this important point. We fully agree with your observation regarding the inclusion of the term “physical activity” in the title and the aim of the study. As suggested, we have removed the term “physical activity” from the title, from the aim, and from the discussion section.
Moreover, as recommended, we have added to the limitations section that the absence of specific assessments of physical fitness (e.g., cardiovascular endurance, muscular strength) represents a limitation of the present study.
We hope that the current version of MS will be suitable for publication.
Reviewer 2 Report
Comments and Suggestions for Authors
Dear authors,
Thank you for the opportunity to review your manuscript "Feasibility and impact of 6-month rowing on physical activity, arm lymphedema, flexibility, and fatigue in breast cancer survivors." Your research addresses an important topic in cancer survivorship and rehabilitation. I found your investigation into sculling as a potential physical activity intervention for breast cancer survivors to be particularly novel and valuable to the field. The manuscript is well-written, methodologically sound, and makes a significant contribution to our understanding of exercise options for breast cancer survivors.
Abstract
The abstract successfully communicates the study's purpose, key methodology elements, main findings on safety and efficacy, and concludes with appropriate implications regarding sculling for breast cancer survivors.
Suggestions for improvement:
- Avoid abbreviations like "BC" and "PA" in abstract
- Condense methodology into a single, comprehensive sentence that includes: study design, participant sample, intervention components, and assessment timepoints
- Include more specific numerical results, particularly for the significant lymphedema improvements and flexibility gains
- Strengthen the conclusion by more explicitly stating results
Introduction
The introduction provides a comprehensive overview of breast cancer epidemiology, treatment side effects, and the role of physical activity in rehabilitation, with a thorough and well-structured literature review on various sports interventions for breast cancer survivors that presents the rationale for investigating rowing activities, progresses logically from general cancer information to specific rowing modalities, and offers strong justification for examining sculling based on its unique biomechanical characteristics.
Suggestions for improvement:
- The research gap could be more explicitly stated - while the authors note "limited attention has been directed toward the potential benefits of sculling for BC survivors," they could more clearly articulate what specific knowledge is missing in the current literature
- The study aims stated in the final paragraph could be more specific regarding expected outcomes
- Consider adding a brief statement about the potential clinical significance of this research to strengthen the rationale
Materials and Methods
The methodology section presents a clear description of the study design, participant selection, intervention protocol, and assessment procedures, with comprehensive inclusion and exclusion criteria, detailed explanation of the 24-week training program components, appropriate measurement tools for all outcomes, and a thorough statistical analysis approach that is well justified for the research questions.
Suggestions for improvement:
- Provide a brief description of how adverse events were defined and recorded during the study, as this would strengthen the safety conclusions
- A short statement about ethical considerations and the participant-informed consent process is needed
- For improved reproducibility, include a few more details about the specific exercises used in the adapted physical activity online classes
- Clarify whether any participant feedback was collected regarding program acceptability, as this would enhance the feasibility assessment
Results
The results section presents findings in a clear, organized manner with appropriate statistical reporting and well-structured tables, providing comprehensive data on participant characteristics, adherence rates, outcome measures across all timepoints, appropriate statistical values including effect sizes, and balanced reporting of both significant findings (arm volume and flexibility improvements) and non-significant results.
Suggestions for improvement:
- Consider including a figure showing the changes in key outcome variables (particularly arm volume and flexibility) over time to enhance visual interpretation of the significant findings
- Clarify whether the improvements in lymphedema were consistent across participants with different cancer stages or surgical interventions, as this would strengthen clinical relevance
- While effect sizes (partial eta squared and Cohen's d) are reported, consider briefly discussing the clinical meaningfulness of these observed changes, particularly for the significant findings in arm volume and flexibility
Discussion
The discussion effectively interprets the study findings in the context of existing literature, addressing each outcome measure and relating the results to previous research on rowing and other physical activities for breast cancer survivors, with thorough comparisons to previous studies, thoughtful analysis of sculling's unique benefits for flexibility, honest acknowledgment of both significant and non-significant findings, good examination of adherence rates for feasibility assessment, and appropriate recognition of study limitations.
Suggestions for improvement:
- Expand on the clinical significance of the lymphedema improvements, particularly the reduction from 15% to 0% of participants with lymphedema, as this represents a meaningful outcome for breast cancer survivors
- Consider briefly discussing potential reasons why fatigue measures did not show significant improvements, which might help inform future research directions
- The discussion of sculling's biomechanical advantages is strong; consider linking these more explicitly to the observed improvements in flexibility
- The recommendations for future research are appropriate; consider also suggesting whether a longer intervention duration might yield different results for outcomes that showed positive trends but didn't reach statistical significance
Conclusion
The conclusion effectively summarizes the key findings of the study, highlighting the safety and potential benefits of sculling for breast cancer survivors, with a clear restatement of the main results, appropriate acknowledgment of the study's novel contribution, realistic assessment of technical challenges, and recognition of participants' ability to master this demanding activity.
Suggestions for improvement:
- Consider adding a brief statement explicitly linking the findings to clinical practice recommendations
- A concise statement about future research directions would strengthen the conclusion
References
The reference list is comprehensive and up-to-date, including relevant literature from both breast cancer research and physical activity/exercise science fields.
Author Response
Thank you for the opportunity to review your manuscript "Feasibility and impact of 6-month rowing on physical activity, arm lymphedema, flexibility, and fatigue in breast cancer survivors." Your research addresses an important topic in cancer survivorship and rehabilitation. I found your investigation into sculling as a potential physical activity intervention for breast cancer survivors to be particularly novel and valuable to the field. The manuscript is well-written, methodologically sound, and makes a significant contribution to our understanding of exercise options for breast cancer survivors.
We sincerely thank the reviewer for the detailed and thoughtful comments provided. The constructive feedback has been extremely valuable and has helped us to improve the clarity, completeness, and overall quality of the manuscript.
R2.1. Abstract
The abstract successfully communicates the study's purpose, key methodology elements, main findings on safety and efficacy, and concludes with appropriate implications regarding sculling for breast cancer survivors.
Suggestions for improvement:
- Avoid abbreviations like "BC" and "PA" in abstract
- Condense methodology into a single, comprehensive sentence that includes: study design, participant sample, intervention components, and assessment timepoints
- Include more specific numerical results, particularly for the significant lymphedema improvements and flexibility gains
- Strengthen the conclusion by more explicitly stating results
Thank you for your positive feedback on the abstract. We have revised it in accordance with the points outlined. Please find below the list of corrections made:
- We have replaced the abbreviation "BC" with the full term "breast cancer" for clarity. The term "physical activity" has been removed from the text entirely.
- We have reformulated the methodology section into a single sentence.
- We have included the specific numerical results for the significant changes observed in the operated limb.
- We have strengthened the conclusion as suggested by more explicitly stating the key results.
R2.2. Introduction
The introduction provides a comprehensive overview of breast cancer epidemiology, treatment side effects, and the role of physical activity in rehabilitation, with a thorough and well-structured literature review on various sports interventions for breast cancer survivors that presents the rationale for investigating rowing activities, progresses logically from general cancer information to specific rowing modalities, and offers strong justification for examining sculling based on its unique biomechanical characteristics.
Suggestions for improvement:
- The research gap could be more explicitly stated - while the authors note "limited attention has been directed toward the potential benefits of sculling for BC survivors," they could more clearly articulate what specific knowledge is missing in the current literature
A clear definition of the specific knowledge missing in the current literature has been provided in the revised manuscript. We inserted the sentence “However, no studies have specifically examined the feasibility, safety, or potential benefits of sculling, such as its effects on lymphedema or fatigue, in BC survivors.” to explicitly highlight the research gap.
- The study aims stated in the final paragraph could be more specific regarding expected outcomes
As recommended, we have revised the final paragraph of the Introduction to clearly specify the expected outcomes of the intervention. We added the sentence:
“Given its biomechanical characteristics, the outdoor setting, and the team-based nature of sculling, we hypothesized that the intervention would be feasible and well-tolerated, and would lead to improvements in flexibility, a reduction in lymphedema, and decreased cancer-related fatigue.”
- Consider adding a brief statement about the potential clinical significance of this research to strengthen the rationale
We have revised the manuscript to include a brief statement emphasizing the potential clinical implication of this study. Specifically, we added the following sentence:
“These attributes and the current lack of evidence highlight the need for further investigation into the role of sculling as a potentially valuable component of survivorship care in BC rehabilitation programs, with the potential to inform clinical exercise guidelines and expand accessible rehabilitation pathways for breast cancer survivors.”
R2.3 Materials and Methods
The methodology section presents a clear description of the study design, participant selection, intervention protocol, and assessment procedures, with comprehensive inclusion and exclusion criteria, detailed explanation of the 24-week training program components, appropriate measurement tools for all outcomes, and a thorough statistical analysis approach that is well justified for the research questions.
Suggestions for improvement:
- Provide a brief description of how adverse events were defined and recorded during the study, as this would strengthen the safety conclusions
We have added a sentence that specifies the definition of adverse events and explains how they were monitored during the study: “Any physical complaints, injuries, or symptoms interfering with session participation were considered adverse events and were monitored weekly through direct feedback from participants to trainers and kinesiologists.”
- A short statement about ethical considerations and the participant-informed consent process is needed
We have already included information regarding ethical considerations and the participant-informed consent process in the manuscript and in the statements section at the end of the manuscript.
- For improved reproducibility, include a few more details about the specific exercises used in the adapted physical activity online classes
We have provided details about the adapted physical activity sessions to improve reproducibility by including the following sentence:
“The sessions comprised a warm-up phase including muscle activation, mobility, and postural control exercises; a workout phase characterized by resistance circuit training; and concluded with a cool-down phase focusing on flexibility and relaxation techniques.”
- Clarify whether any participant feedback was collected regarding program acceptability, as this would enhance the feasibility assessment
We have added the following sentence to address this point:
“Additionally, participant feedback on program acceptability was collected regularly through informal discussions with trainers and kinesiologists to better assess the feasibility and overall participant experience of the intervention.”
R2.4 Results
The results section presents findings in a clear, organized manner with appropriate statistical reporting and well-structured tables, providing comprehensive data on participant characteristics, adherence rates, outcome measures across all timepoints, appropriate statistical values including effect sizes, and balanced reporting of both significant findings (arm volume and flexibility improvements) and non-significant results.
Suggestions for improvement:
- Consider including a figure showing the changes in key outcome variables (particularly arm volume and flexibility) over time to enhance visual interpretation of the significant findings.
We have addressed this comment by including two figures that graphically represent changes in arm volume and flexibility over time, to enhance the visual interpretation of the main findings.
- Clarify whether the improvements in lymphedema were consistent across participants with different cancer stages or surgical interventions, as this would strengthen clinical relevance
Yes, the changes occurred consistently across the sample, regardless of cancer stage or type of surgical intervention. We have included a sentence clarifying this point.
- While effect sizes (partial eta squared and Cohen's d) are reported, consider briefly discussing the clinical meaningfulness of these observed changes, particularly for the significant findings in arm volume and flexibility
We have included the following sentence discussing the clinical meaningfulness of these observed changes, highlighting their potential impact despite small effect sizes:
“Despite small effect sizes, the improvements in lymphedema and flexibility may have meaningful implications for functional mobility and the performance of daily activities in BC survivors.”
R2.5 Discussion
The discussion effectively interprets the study findings in the context of existing literature, addressing each outcome measure and relating the results to previous research on rowing and other physical activities for breast cancer survivors, with thorough comparisons to previous studies, thoughtful analysis of sculling's unique benefits for flexibility, honest acknowledgment of both significant and non-significant findings, good examination of adherence rates for feasibility assessment, and appropriate recognition of study limitations.
Suggestions for improvement:
- Expand on the clinical significance of the lymphedema improvements, particularly the reduction from 15% to 0% of participants with lymphedema, as this represents a meaningful outcome for breast cancer survivors
We have included the following sentence as suggested by the reviewer to explain the clinical significance of the lymphedema improvements:
“Considering the physical and psychological burden associated with lymphedema, as well as the risk of exacerbation through physical exercise, the observed reduction in lymphedema prevalence among participants represents a clinically meaningful and important outcome for BC survivors.”
- Consider briefly discussing potential reasons why fatigue measures did not show significant improvements, which might help inform future research directions
We included a possible explanation in the discussion section. Specifically, we noted that over 50% of participants were undergoing hormonal therapy, which is commonly associated with persistent fatigue and may have limited the intervention’s effectiveness in reducing fatigue levels.
- The discussion of sculling's biomechanical advantages is strong; consider linking these more explicitly to the observed improvements in flexibility
We have added a sentence that more explicitly explains the observed improvements in flexibility.
- The recommendations for future research are appropriate; consider also suggesting whether a longer intervention duration might yield different results for outcomes that showed positive trends but didn't reach statistical significance
This suggestion has been considered in the conclusion section.
R2.6 Conclusion
The conclusion effectively summarizes the key findings of the study, highlighting the safety and potential benefits of sculling for breast cancer survivors, with a clear restatement of the main results, appropriate acknowledgment of the study's novel contribution, realistic assessment of technical challenges, and recognition of participants' ability to master this demanding activity.
Suggestions for improvement:
- Consider adding a brief statement explicitly linking the findings to clinical practice recommendations
We have added a brief statement linking the findings to clinical practice recommendations.
- A concise statement about future research directions would strengthen the conclusion
We have included a concise statement about future research directions in the conclusion section, as suggested by the reviewer.
R2.7 References
The reference list is comprehensive and up-to-date, including relevant literature from both breast cancer research and physical activity/exercise science fields.
Thank you very much for your positive feedback regarding the reference list.
Reviewer 3 Report
Comments and Suggestions for Authors
Please see the attached document.

Author Response
I am glad that I had the opportunity to review this study. The topic you are covering is very interesting from a scientific point of view, but also from a practical aspect in BC rehabilitation. Below I present a few notes that I believe can contribute to your work. All of the above notes can be included in the article relatively easily.
Thank you very much for your thoughtful and encouraging comments. We truly appreciate the time you took to review our study and your valuable insights.
- Ln 244 - Please attach a reference to the applied questionnaire.
We have added the requested reference for the applied questionnaire as suggested.
- Ln 254 – I suggest that you present the results of the normality and sphericity test in table 2;
Thank you for your valuable suggestions. We have included the results of the sphericity tests in Tables 2 and 3, along with the applied corrections when sphericity was violated.
- If and when this principle of normality and sphericity is violated, state it in the results section
Thank you for this indication. We believe that explicitly restating these details in the results text might be redundant, as the information is clearly presented in the tables.
- Ln 275 – table 2 is difficult to read due to condensed columns; enlarge the table
Done.
- Consider showing the results graphically.
We have included two figures illustrating the changes in arm volume and flexibility, thereby improving the visual interpretation of the key outcomes.
- Ln 338 – please further clarify the modalities of work in your study that could have resulted in such findings
We have added a clarifying sentence to better describe the training modalities.
- Ln 431 - I think it would be useful to provide direct recommendations in this section since the results of your study seem promising
As recommended, we have included a direct recommendation in the conclusion section.